# Late Chunking: Contextual Chunk Embeddings Using Long-Context Embedding Models

## Abstract

Many use cases require retrieving smaller portions of text, and dense vector-based retrieval systems often perform better with shorter text segments, as the semantics are less likely to be "over-compressed" in the embeddings. Consequently, practitioners often split text documents into smaller chunks and encode them separately. However, chunk embeddings created in this way can lose contextual information from surrounding chunks, resulting in sub-optimal representations. In this paper, we introduce a novel method called "late chunking", which leverages long context embedding models to first embed all tokens of the long text, with chunking applied *after* the transformer model and just before mean pooling - hence the term "late" in its naming. The resulting chunk embeddings capture the full contextual information, leading to superior results across various retrieval tasks. The method is generic enough to be applied to a wide range of long-context embedding models and works without additional training. To further increase the effectiveness of late chunking, we propose a dedicated fine-tuning approach for embedding models.

## 1 Introduction

Neural information retrieval (IR) relies on text embedding models (Reimers & Gurevych, 2019) that are primarily based on the transformer architecture (Devlin et al., 2019) and have been pre-trained using very large text corpora. These models capture important elements of texts' semantics in the form of dense vectors whose spatial relations - particularly cosine distance - are good proxies for text similarity and relevancy. For many neural IR use cases like the well-known RAG (Retrieval Augmented Generation) approach (Lewis et al., 2020), applications require splitting documents into limited-size text chunks, and storing them and their vector embeddings in a database. At run-time, neural IR techniques are used to retrieve chunks of text relevant to a user's requests, which are, in the case of RAG, presented to an LLM as a basis for synthesizing a response. Furthermore, many other applications require processing small text segments (Salton et al., 1993; Demszky et al., 2020), and therefore rely on chunking, e.g., to quickly navigate a user to the relevant passage in a document Callan (1994).

Moreover, while long context embedding models can improve retrieval performance on long texts (Günther et al., 2023), they still perform better on short texts (Zhou et al., 2024). As a result, chunking generally improves retrieval, even with models that support long contexts. (See Appendix A.1.) However, long-distance semantic dependencies – when the relevant information to interpret one chunk of text is located in one or more other chunks – reduce the effectiveness of this search strategy. Figure 1 displays a Wikipedia article[1] that is split into chunks of sentences. One can see that phrases like "its" and "the city" referencing "Berlin" which is mentioned only in the first sentence, e.g., it is harder for the embedding model to link it to the respective entity to produce a high-quality representation.

To overcome this limitation, we introduce a novel technique called *late chunking*. This method leverages the long text embedding capabilities of recently published open source models (Günther et al., 2023; Nussbaum et al., 2024) to, first, encode all tokens of an entire document with their full in-document context into a sequence of token embeddings, and then break this sequence up into chunks, which receive embeddings via mean pooling of their token embeddings. This way, chunk embeddings include relevant semantic information derived from their place in the whole text.

---

[1] `https://en.wikipedia.org/wiki/Berlin` (Access 09-30-2024)

Figure 1: An illustration of the lost context problem. A Wikipedia article about Berlin is split into chunks. One can see that phrases like "its" and "the city" reference "Berlin," which is mentioned only in the first sentence. This makes it harder for the embedding model to link these references to the correct entity, thereby producing a lower-quality vector representation.

Table 1: Comparing the embedding of the term "Berlin" to various sentences from the article about Berlin using cosine similarity. The column "Sim. naive" shows the similarity values between the query embedding of "Berlin" and the embeddings using naive chunking, while "Sim. late" represents the results with the late chunking method.

| Text | Sim. Naive | Sim. Late |
|---|---|---|
| Berlin is the capital and largest city of Germany, both by area and by population. | 0.8486 | 0.8495 |
| Its more than 3.85 million inhabitants make it the European Union's most populous city, as measured by population within city limits. | 0.7084 | 0.8249 |
| The city is also one of the states of Germany, and is the third smallest state in the country in terms of area. | 0.7535 | 0.8498 |

As an example of how late chunking works, we encode the texts in Figure 1 with a long-context embedding model, `jina-embeddings-v2-small`, using both naive and late chunking methods. We then calculate the similarity of the resulting embeddings to the embedding of the word "Berlin". Table 1 shows that, with naive chunking, texts that do not contain the word "Berlin" have low similarity scores, even though both sentences, in context, refer to the city of Berlin. With late chunking, you can see that the similarity scores are much higher. The late chunking strategy has encoded "Berlin" into the embeddings of "Its" and "the city" because it sees them in their context before chunking the text.

Late chunking is an architectural change that can be implemented in any long-context text embedding model that uses mean pooling with any chunking technique and does not require additional model training. It leads to superior results compared to naive chunking across a wide range of retrieval benchmarks. To demonstrate the replicability of our results, we are publishing the code via GitHub [2]. In particular, we make the following contributions:

- **Late Chunking:** We describe our novel late chunking technique in Section 3 and demonstrate that it leads to superior results compared to naive chunking across a wide range of retrieval benchmarks.

- **Extended Algorithm for Long Documents:** For encoding long documents with more tokens than long-context embedding models can handle, we propose a long late chunking approach (see Section 3.1) and prove its effectiveness in Section 4.3.

---

[2]Link omitted for anonymization

- **Training for Late Chunking:** While late chunking does not require additional training, we propose a novel training method to further enhance retrieval accuracy when using it (see Section 3.2). We conduct an evaluation to show its advantage over comparable contrastive training in Section 4.4.

- **Comprehensive Evaluation:** We conduct a comprehensive empirical evaluation to identify scenarios where late chunking performs superior to naive chunking and scenarios where the standard method yields comparable or superior results (see Sections 4.1 and 4.2).

## 2 RELATED WORK

Most modern text embedding models are trained on transformer-based architectures (Devlin et al., 2019) using the training method proposed by Reimers & Gurevych (2019). In general, the model is equipped with a pooling operator which converts the token embeddings produced by the transformer into a single vector representation. Mean pooling is especially popular, as Reimers & Gurevych (2019) conduct experiments in which mean pooling shows the best performance among other methods. While the original transformer uses absolute positional encodings, methods that encode relative positions like AliBi (Press et al., 2022) and RoPE (Su et al., 2024) allow effective training of embedding models with larger context lengths (Günther et al., 2023; Nussbaum et al., 2024).

To address the limited context length and overcome practical issues of handling embeddings of long texts, chunking text before embedding it has become common practice. While simple chunking methods use a fixed token length (Lewis et al., 2020) or split text into units like sentences or paragraphs, more sophisticated methods like semantic chunking (Kamradt, 2024) use the similarity of embedding vectors of neighboring sentences to find optimal spans for chunking.

To prevent the problem of missing context information various approaches have been proposed that augment the text of the chunks. For instance, practitioners divide text into overlapping chunks (Safjan, 2023), meaning that the end of one chunk shares some tokens with the beginning of the next chunk. During the development of this paper, a blog post (Anthropic, 2024) introduced an alternative approach for producing contextualized chunk embeddings using an additional large language model (LLM). The LLM receives as input the whole document and the target chunk to produce text for augmenting the chunk text with relevant context information before passing it to the embedding model. This is however computationally more expensive, as LLMs are typically much larger than embedding models or even require paid access to LLM APIs. Similarly, Luo et al. (2024) extract propositions for each paragraph using an additional language model. However, each paragraph is processed independently, which might result in losing context across paragraphs. Moreover, this approach cannot be used with any technique for segmenting the text (e.g. fixed-size, sentence-based, semantic chunking, ...) but is restricted to the texts produced by the language model.

Another branch of research proposes embedding models that encode and index an embedding for each token. Models like ColBERT (Khattab & Zaharia, 2020; Jha et al., 2024) use a method called "late interaction", which compares each token embedding of the query with each token embedding of the document and can compute more accurate relevance scores in this way. However, in contrast to our proposed late chunking method, this leads to more computational effort during the vector search.

Furthermore, Chen et al. (2024) aims to produce contextualized embedding representations by training an embedding model specifically to produce contextualized embedding representations of sentences.

## 3 METHOD

*Late chunking* is a strategy for taking advantage of the difference in size between the long context input windows of recent embedding models and the relatively small size of optimal text chunks for most applications. These models support much longer input texts, for example, 8192 tokens for `jina-embeddings-v2-small` – roughly ten pages of standard text – while optimal chunk sizes are typically much smaller, e.g., the size of a paragraph. The reasons can be manifold, one being that LLMs get more inefficient when providing longer context, and a single short embedding vector only has a limited capacity to represent information.

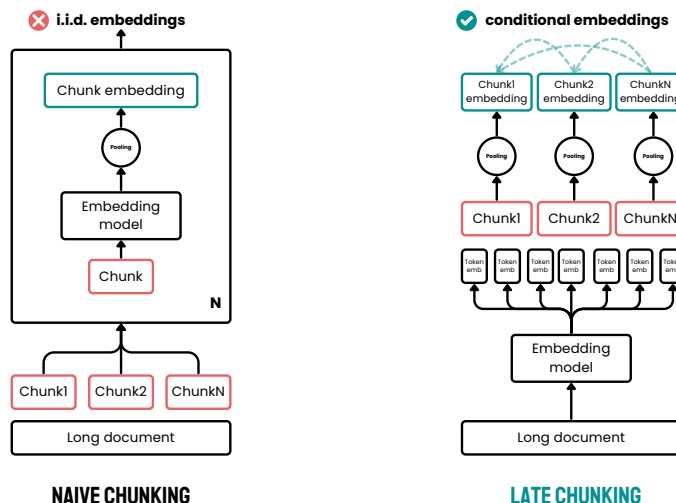

Figure 2: Overview of the naive chunking strategy (left) and the late chunking strategy (right). In late chunking, the transformer processes the entire text first, allowing chunk embeddings to capture context from the whole text, unlike the naive approach which first splits the text into sub-strings which are passed as independent units to the model.

The naive chunking approach (left side in Figure 2) chunks texts before processing them, using sentences or paragraphs, and then applies an embedding model to the resulting chunks. Contrastively, late chunking, as described in Algorithm 1, first tokenizes the entire text, or the largest part of it possible (line 2), and applies the transformer part from the embedding model on it (line 3). This generates a sequence of vector representations $\vartheta_1, \ldots, \vartheta_m$ for each token that encompass textual information from the entire text. To generate a single embedding for a text, many embedding models apply mean pooling to these token representations to output a single vector. Late chunking instead applies mean pooling to smaller segments of this sequence of token vectors, producing embeddings for each chunk that take into account the entire text. It is important to highlight that late chunking still requires boundary cues that are derived from the chunks determined by a chunking algorithm, but these cues are used only *after* obtaining the token-level-embeddings - hence the term *late* in its naming. Chunking algorithms usually chunk text into sequences of characters. For late chunking, boundary cues corresponding to a sequence of tokens are necessary. Accordingly, Lines 5-13 of the algorithm translate the chunk definition into boundary cues that are used by the pooling step in the lines14-16.

### 3.1 LONG LATE CHUNKING

Although many embedding models offer a high enough context length to encode a large amount of text at once, the context length might still not be sufficient to encode very large documents in one step. Moreover, the memory required for the encoding increases exponentially with an increasing number of tokens so that encoding all tokens at once becomes infeasible. To solve this problem, we propose using long late chunking as described in Algorithm 2. Thereby, the text is split into larger macro chunks of $l_{max}$ tokens that encompass multiple smaller chunks. Each macro chunk is processed separately by the LATECHUNKING method. To avoid missing context, macro chunks are augmented with a certain number of tokens $\omega$ that overlap with the next chunks. Those additional tokens serve as supplementary context information during late chunking.

### 3.2 TRAINING METHOD

While late chunking works without further training, models that are trained with mean pooling to create a single embedding representation of a longer text might not be well-suited to encode chunks of token embeddings containing additional information from surrounding tokens. Therefore, we

---

**Algorithm 1** Late Chunking

**Inputs:** Text $T$, Chunking Strategy $S$
**Outputs:** Chunk Embeddings $e_1, \ldots, e_n$

1: $(c_1, \ldots, c_n) \leftarrow \text{Chunker}(T, S)$
2: $(\tau_1, \ldots, \tau_m), (o_1, \ldots, o_m) \leftarrow \text{Tokenizer}(T)$      $\triangleright$ $\tau_i$ is a token ID, $o_i$ its character length
3: $(\vartheta_1, \ldots, \vartheta_m) \leftarrow \text{Model}(\tau_1, \ldots, \tau_m)$      $\triangleright$ Calculate token embeddings $\vartheta_1, \ldots, \vartheta_m$
4: $o_{\text{chunk}} \leftarrow 0, j \leftarrow 1, cue_{start} \leftarrow 1, cues \leftarrow [\,]$

5: **for** $i \in \{1, \ldots, m\}$ **do**      $\triangleright$ For each token
6:      $o_{\text{chunk}} \leftarrow o_{\text{chunk}} + o_i$
7:      **if** $o_{\text{chunk}} \geq |c_j|$ **then**      $\triangleright$ When the current chunk size is reached, save positions
8:          $cue_{end} \leftarrow i$
9:          $cues \leftarrow cues \oplus (cue_{start}, cue_{end})$
10:         $j \leftarrow (j + 1), cue_{start} \leftarrow (i + 1)$
11:         $o_{\text{chunk}} \leftarrow 0$
12:      **end if**
13: **end for**

14: **for** $(cue_{start}, cue_{end})_i \in cues$ **do**      $\triangleright$ Pool token embeddings according to $cue$ positions
15:      $e_i \leftarrow \left( \sum_{j=cue_{start}}^{cue_{end}} \vartheta_j \right) / ((cue_{end} + 1) - cue_{start})$
16: **end for**

---

**Algorithm 2** Long Late Chunking

**Inputs:** Text $T$, Chunking Strategy $S$, Maximum Token Length $l_{max}$, Overlap Length $\omega$
**Outputs:** Chunk Embeddings $E = (e_1, e_2, \ldots, e_n)$

1: $(c_1, \ldots, c_n) \leftarrow \text{Chunker}(T, S)$
2: $(\tau_1, \tau_2, \ldots, \tau_m), (o_1, o_2, \ldots, o_m) \leftarrow \text{Tokenizer}(T)$      $\triangleright$ $\tau_i$ is a token ID, $o_i$ its character length

3: **if** $m < l_{max}$ **then**      $\triangleright$ If the number of tokens is already small, do regular late chunking
4:      **return** LateChunking$(T, S)$
5: **end if**

6: $i_{\text{end}} \leftarrow 1, \ \ \text{embeddings} \leftarrow [\,]$
7: **while** $i_{\text{end}} < m$ **do**
8:      $i_{\text{start}} \leftarrow \max(i_{\text{end}} - \omega, 1)$      $\triangleright$ Update token positions with overlap
9:      $i_{\text{end}} \leftarrow \min(i_{\text{start}} + l_{max}, m)$
10:     $(\vartheta_{i_{\text{start}}}, \ldots, \vartheta_{i_{\text{end}}}) \leftarrow \text{Model}(\tau_{i_{\text{start}}}, \ldots, \tau_{i_{\text{end}}})$      $\triangleright$ Calculate token embeddings
11:     **if** $i_{\text{start}} = 1$ **then**
12:         $\text{embeddings} \leftarrow \text{embeddings} \oplus (\vartheta_{i_{\text{start}}}, \ldots, \vartheta_{i_{\text{end}}})$
13:     **else**
14:         $\text{embeddings} \leftarrow \text{embeddings} \oplus (\vartheta_{i_{\text{start}}+\omega}, \ldots, \vartheta_{i_{\text{end}}})$
15:     **end if**
16: **end while**

17: Carry out steps 4 to 16 of Algorithm 1 with augmented token embeddings $\vartheta_1, \ldots, \vartheta_m$.

---

propose a modified text embedding training method, which uses a technique that we call "span

pooling" to train the model to encode specifically the relevant information contained in an annotated text span into its token embeddings.

**Training Data:** To conduct the training, we prepare training data which consist of tuples $(q, d, \langle start, end \rangle)$ of two text values: a query $q$ and a relevant document $d$, with additional annotation of the relevant span in the document $\langle start, end \rangle$ that contains the answer.

**Training Process:** The fine-tuning procedure itself follows the pair training stage described in Günther et al. (2023), where the model is trained on text pairs using the InfoNCE loss function (van den Oord et al., 2018) which is defined on a batch $B = ((x_1, y_1), \ldots, (x_k, y_k))$ of $k$ pairs and the cosine similarity function $s$:

$$\mathcal{L}_{\text{NCE}}(B) := - \sum_{(x_i, y_i) \in B} \ln \frac{e^{s(x_i, y_i)/\tau}}{\sum\limits_{i'=1}^{k} e^{s(x_i, y_{i'})/\tau}} \tag{1}$$

Here, the query vectors $x_i$ are obtained by applying the embedding model to the query text $q_i$ in the usual way. For the document embeddings $y_i$, the set of token embeddings $\vartheta_{i,1}, \ldots, \vartheta_{i,n}$ is obtained by applying the model on the documents $d_i$, and executing the mean pooling operation only to the token embeddings within the span $\langle start, end \rangle$, hence the term "span pooling".

As proposed by Günther et al. (2023), we use a bi-directional version of the loss $\mathcal{L}_{\text{pairs}}$, where $B^\dagger = ((y_1, x_1), \ldots, (y_k, x_k))$ is obtained from $B$ by swapping the order of pairs:

$$\mathcal{L}_{\text{pairs}}(B) := \mathcal{L}_{\text{NCE}}(B) + \mathcal{L}_{\text{NCE}}(B^\dagger) \tag{2}$$

A description of the datasets, hyperparameters of the training and the evaluation results can be found in Section 4.4.

## 4 EVALUATION

First, we evaluate late chunking on a variety of models, chunking methods, and retrieval datasets to show its effectiveness in Section 4.1. Section 4.2 investigate the influence of the chunking size and also identifies scenarios where late chunking works optimally, as well as limitations of the method. The long late chunking method is evaluated on datasets with long documents in Section 4.3. The proposed span pooling method for training is evaluated in Section 4.4. Finally, we also conduct a small-scale evaluation to compare late chunking to the LLM-based contextual embedding technique in Section 4.5.

### 4.1 EVALUATION ON RETRIEVAL TASKS

To test the effectiveness of late chunking, we apply our technique to the smaller retrieval tasks of the BeIR benchmark (Thakur et al., 2021). We restrict the evaluation on the smaller datasets, as splitting documents into smaller chunks increases the computational effort of the evaluation, which makes a comprehensive evaluation on different models, tasks, and chunking techniques infeasible.

Those retrieval tasks consist of a query set, a corpus of text documents, and a QRels file that stores information about the IDs of documents that are relevant for each query. To identify the relevant documents of a query, one can chunk the documents, encode and store them into an embedding index, and determine for each query embedding the chunks corresponding to the k-nearest-neighbors (kNN) of their normalized vector representations. As each chunk corresponds to a document, one can convert the kNN ranking of chunks into a kNN ranking of documents (for documents occurring multiple times in the ranking, only the first occurrence is retained). After that, one can compare the resulting ranking with the ranking corresponding to the ground-truth QRels file and calculate retrieval metrics like nDCG@10.

We run this evaluation for the BeIR datasets with naive chunking, our novel late chunking method, and also report the score obtained without chunking. Both naive chunking and late chunking are evaluated with different chunking techniques, we use:

Table 2: Evaluation of different chunking methods on retrieval tasks. Scores are reported as nDCG@10 [%] Models: `jina-embeddings-v2-small` (J2s), `jina-embeddings-v3`(J3), nomic-embed-text-v1 (Nom).

| | SciFact | | | NFCorpus | | | FiQA | | | TRECCOVID | | | AVG |
|---|---|---|---|---|---|---|---|---|---|---|---|---|---|
| | J2s | J3 | Nom | J2s | J3 | Nom | J2s | J3 | Nom | J2s | J3 | Nom | |
| **Fixed-Size Boundaries** (256 Tokens per Chunk) | | | | | | | | | | | | | |
| **Naive** | 64.2 | 71.8 | **70.7** | 23.5 | 35.6 | 35.3 | 33.3 | 46.3 | 37.0 | 63.4 | 73.0 | 72.9 | 52.2 |
| **Late** | **66.1** | **73.2** | 70.6 | **30.0** | **36.7** | 35.3 | **33.8** | **47.6** | **38.3** | **64.7** | **77.2** | **75.0** | **54.0** |
| **Sentence Boundaries** (5 Sentences per Chunk) | | | | | | | | | | | | | |
| **Naive** | 64.7 | 71.4 | 71.3 | 28.3 | 35.8 | 34.7 | 30.4 | 43.7 | 35.1 | 66.5 | 72.4 | 74.2 | 52.4 |
| **Late** | **65.2** | **73.2** | **71.4** | **30.0** | **36.6** | **35.5** | **33.9** | **48.0** | **37.7** | **66.6** | **76.5** | **76.8** | **54.3** |
| **Semantic Sentence Boundaries** | | | | | | | | | | | | | |
| **Naive** | 64.3 | 71.2 | 70.4 | 27.4 | 36.1 | 35.3 | 30.3 | 44.0 | 34.8 | 66.2 | 74.7 | 74.3 | 52.4 |
| **Late** | **65.0** | **72.4** | **70.5** | **29.3** | **36.6** | 35.3 | **33.7** | **47.6** | **36.9** | **66.3** | **76.2** | **76.1** | **53.8** |

- **Fixed-Size Boundaries:** Each chunk has the same number of tokens (256 in this experiment).
- **Sentence Boundaries:** Each chunk has the same number of sentences (5 in this experiment).
- **Semantic Sentence Boundaries:** Each chunk corresponds to multiple sentences. Sentences with high embedding similarity (we use jina-embeddings-v2-small-en) are combined in the same chunk. We use the semantic chunking implementation from llama-index[3] with the default parameters.

We evaluate three embedding models: `jina-embeddings-v2-small` (Günther et al., 2023), `jina-embeddings-v3` (Sturua et al., 2024), and `nomic-embed-text-v1` (Nussbaum et al., 2024).

**Dealing with Non-Context Tokens:** Not all tokens correspond to characters in the original string. For instance, the tokenizers of all models add a [CLS] token at the beginning and append a [SEP] token at the end of the text. Additionally, `jina-embeddings-v3` and `nomic-embed-text-v1` prepend an instruction to the string for distinguishing queries and documents. During late chunking, we include all embeddings of prepended tokens in the mean pooling of the first chunk and all embeddings of appended tokens to the last chunk.

We present the evaluation results in Table 2. When comparing the results for the different chunking methods, we observe that replacing naive methods with their late chunking counterparts almost always yields better performance. Averaging results across three models and four datasets, we find a 3.63% relative improvement (1.9% absolute) from naive chunking with sentence boundaries to late chunking using sentence boundaries, a 3.46% improvement (1.8% absolute) from naive chunking to late chunking using fixed-size boundaries, and a 2.70% improvement (1.5% absolute) from naive chunking to late chunking when using semantic sentence boundaries. In all experiments the chunks are non-overlapping, however, additional results demonstrated in appendix A.2 show that overlapping the chunks generally neither improves nor harms the retrieval performance. These findings demonstrate that the late chunking technique effectively and consistently enhances overall performance.

## 4.2 INFLUENCE OF THE CHUNKING SIZE

The following experiment investigates the influence of the chunk size on the performance of naive and late chunking. For this case, we mainly evaluate the model on retrieval tasks with long doc-

---

[3]`https://docs.llamaindex.ai/en/stable/examples/node_parsers/semantic_chunking/`

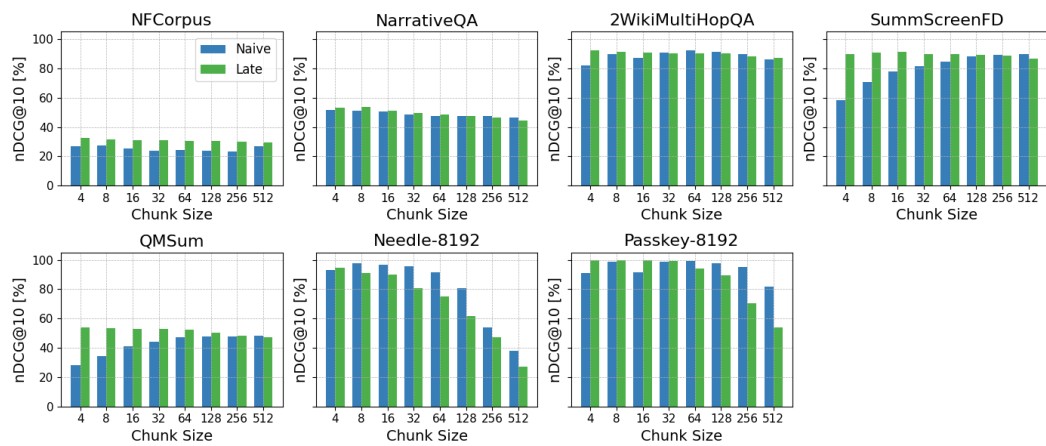

Figure 3: Retrieval Results with Different Chunk Sizes

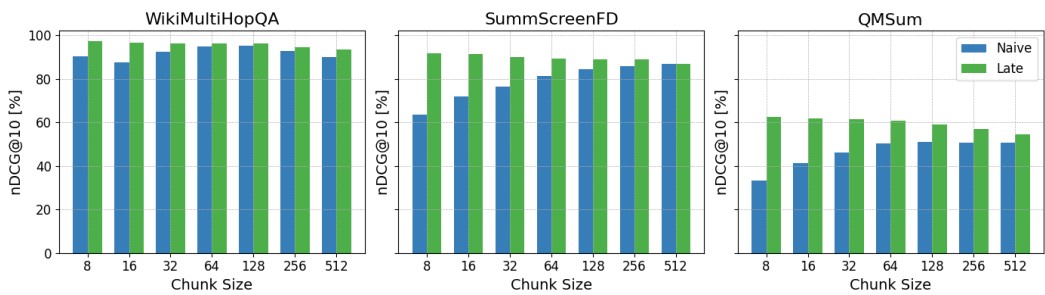

Figure 4: Retrieval Results with Long Late Chunking for Different Chunk Sizes

uments. While most retrieval tasks contain relatively short texts, we only select the NFCorpus dataset from BeIR (which contains comparable long text documents) and also use the datasets from the LongEmbed benchmark (Zhu et al., 2024), which contains retrieval datasets constructed from reading comprehension benchmarks as well as synthetic datasets of long documents. As many documents are longer than 8192 tokens, we truncate the texts at 8192 tokens before the evaluation. We use the chunking method with fixed-size boundaries with different numbers of tokens and evaluate naive and late chunking with `jina-embeddings-v2-small` using the same retrieval evaluation method as described in Section 4.1. The results in Figure 3 show that late chunking performs better than naive chunking, specifically for small chunk sizes. For NFCorpus, late chunking performs consistently better, while for some of the reading comprehension tasks, naive chunking works better when using large chunks. This may be due to some of the reading comprehension datasets requiring finding a specific sentence or phrase embedded into a relatively unrelated textual context instead of finding a whole document. Specifically, the two synthetic datasets Needle-8192 and Passkey-8192 are constructed by placing short relevant information into a document of unrelated text. In this case, late chunking is not useful, as the additional context from the document is totally irrelevant.

## 4.3 EVALUATION OF LONG LATE CHUNKING

To evaluate long late chunking, we select three of the non-synthetic reading comprehension datasets, as none of the BeIR datasets contain a significant amount of text values with more than 8192 tokens. We use the same evaluation method as described in Section 4.2 but do not truncate this time. Figure 4 shows that late chunking with the long late chunking method achieves superior results in comparison to naive chunking. Compared to the experiment of Section 4.2, the nDCG scores are higher, as

Table 3: Evaluation results (nDCG@10 [%]) on chunked evaluation tasks when training with span pooling and mean pooling, with a fixed chunk size of 64 tokens and late chunking during inference.

| Model | Pooling (During Training) | Training Data | Sci-Fact | Narrative-QA | NF-Corpus | TREC-COV | FiQA |
|-------|---------------------------|---------------|----------|--------------|-----------|----------|------|
| J3 | Span-Based | TriviaQA&FEVER | **72.61** | 44.01 | **36.80** | **77.59** | **48.22** |
|    |            | TriviaQA       | 72.28 | **44.94** | 36.69 | 77.39 | 47.99 |
| J3 | Mean | TriviaQA&FEVER | 72.59 | 43.83 | 36.77 | 77.21 | 47.40 |
|    |      | TriviaQA       | 72.56 | 44.86 | 36.78 | 77.36 | 47.35 |
| J2s | Span-Based | TriviaQA&FEVER | **65.20** | 47.29 | 29.96 | **65.18** | **34.52** |
|     |            | TriviaQA       | 65.43 | **47.76** | **30.04** | 64.95 | 34.29 |
| J2s | Mean | TriviaQA&FEVER | 64.77 | 47.31 | 29.70 | 64.73 | 33.87 |
|     |      | TriviaQA       | 65.18 | 47.45 | 29.76 | 64.86 | 33.82 |

truncation in the last experiment could lead to information loss. Long late chunking solves this problem.

## 4.4 EVALUATION OF TRAINING METHOD

Table 3 captures the results from our training experiments. The experiments include running both span-based and regular mean pooling training methods on the `jina-embeddings-v3` and `jina-embeddings-v2-small-en` long context embedding models in order to see whether the proposed training method achieves performance gains in combination with late chunking. To evaluate the models after the training we use the same procedure as in Section 4.1. For chunking, we used fixed-size boundaries (64 tokens). For the `jina-embeddings-v3` model, we fine-tune only the retrieval adapters, following the same hyperparameter settings of Sturua et al. (2024), however with an increased batch size of 512 and training for only 500 steps. The hyperparameters for the fine-tuning of `jina-embeddings-v2-small-en` model are analogous to those detailed in Günther et al. (2023).

For the span-based training method, we prepare two datasets into the format described in Section 3.2 and make these publicly available on HuggingFace [4]. These two datasets are FEVER (Thorne et al., 2018) and TriviaQA (Joshi et al., 2017), which are well-suited for this experiment as they contain annotations of where the relevant text can be found in the documents respectively. In the FEVER dataset, these spans take the shape of sentence number annotations, while for TriviaQA the annotations are usually a name, place, or date in the form of a short phrase. For FEVER, we only include pairs where the document provides supporting evidence for the claim. When multiple spans are annotated in these datasets, we select only the span, which occurs earliest in the document.

Across the datasets and models, span pooling and mean pooling during training deliver relatively similar results, with span pooling consistently achieving a small improvement. The training dataset selection also has a small effect on the performance, thus resulting in slightly higher results for NarrativeQA when only training on TriviaQA, which is likely due to an overlap of domain and phrasing of query-document pairs of the task and training data.

While the span pooling method for training shows promise, the training dataset diversity is quite limited, as both training datasets are sourced from Wikipedia documents. The summed dataset encompasses only ∼470k pairs in total for training, which may additionally limit the potential performance gains. It may be possible to achieve higher performance with a larger quantity and more diverse set of training data.

## 4.5 COMPARISON TO CONTEXTUAL EMBEDDING

We conduct a small-scale experiment to compare late chunking to the LLM-based contextual embedding approach published in a blog post by Anthropic (2024) mentioned in the related

---

[4]Link omitted for anonymization

Table 4: Cosine similarity scores using naive chunking, late chunking, and contextual embedding.

| Chunk | Similarity Late Chunking | Similarity Contextual Embedding | Similarity Naive Chunking |
|---|---|---|---|
| The recent SEC filing provided insights into ACME Corp's performance for Q2 2023. | 0.8305 | 0.8069 | **0.8505** |
| **It highlighted a 3% revenue growth over the previous quarter.** | **0.8516** | **0.8590** | 0.6343 |
| The company, which had a revenue of $314 million in the prior quarter, showed steady progress. | 0.8424 | 0.8546 | 0.6169 |
| They attributed this growth to strategic initiatives and operational efficiencies. | 0.7997 | 0.8234 | 0.5191 |
| The report emphasized the company's resilience and ability to navigate market challenges, reflecting positively on their financial health and future prospects. | 0.8022 | 0.8061 | 0.6007 |

work Section 2. Given the chunks obtained from a fictional financial document shown in Table 4 and the query "What is ACME Corp's revenue growth for Q2 2023?", the goal is to identify the relevant chunk. The relevant chunk in this example, "It highlighted a 3% revenue growth over the previous quarter.", however, misses the company's name, which is necessary to determine its relevancy. We implement the method described in the blog post that uses the `claude-3-haiku-20240307` model to select relevant contextual information from the whole text and add it to the beginning of each text chunk. Then, we encode the query and the augmented chunks with `jinaai/jina-embeddings-v2-small-en` to calculate their cosine similarity. Table 4 captures the similarity values and compares them to those obtained from the chunks with late and naive chunking. One can see that both the contextual embedding method and late chunking produce the highest similarity value for the relevant chunk. In contrast, native chunking leads to a much smaller similarity score that is lower than the similarity to other chunks. Furthermore, one can see that contextual embedding and late chunking produce similarity scores that are close to each other across all chunks, with late chunking having the advantage that it does not require using an additional large language model.

## 5 CONCLUSION

In this paper, we present a novel approach for encoding text chunks with embedding models called *late chunking*. We demonstrate how it can resolve context dependency problems and show that it improves text embeddings across a wide range of retrieval tasks. For handling situations in which the maximum context length of the model is not sufficient, we present a long late chunking approach to effectively solve this problem. Late chunking requires no additional training and is applicable to a wide range of embedding models. Furthermore, we demonstrate that additional training with a custom method can further enhance its performance on retrieval tasks.

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

# A APPENDIX

## A.1 LIMITATIONS OF LONG-CONTEXT EMBEDDING MODELS

Table 5: Influence of truncation and chunk size on retrieval with long texts. Truncation is done *before* chunking.
Model: `jina-embeddings-v2-small`, naive (not late) chunking with fixed-size strategy.

| NarrativeQA Dataset | | | |
|---|---|---|---|
| **Max. Length (Trucation)** | **Chunk-Size** | **Chunking** | **nDCG@10** |
| 192 | 192 | × | 20.26 |
| 8192 | 8192 | × | 32.73 |
| 8192 | 128 | ✓ | 46.28 |
| 8192 | 512 | ✓ | **47.63** |
| **2WikiMultiHopQA Dataset** | | | |
| **Max. Length (Trucation)** | **Chunk-Size** | **Chunking** | **nDCG@10** |
| 192 | 192 | × | 48.86 |
| 8192 | 8192 | × | 70.32 |
| 8192 | 128 | ✓ | **91.36** |
| 8192 | 512 | ✓ | 86.3 |
| **SummScreenFD Dataset** | | | |
| **Max. Length (Trucation)** | **Chunk-Size** | **Chunking** | **nDCG@10** |
| 192 | 192 | × | 52.89 |
| 8192 | 8192 | × | **91.24** |
| 8192 | 128 | ✓ | 88.21 |
| 8192 | 512 | ✓ | 89.71 |
| **QMSum Dataset** | | | |
| **Max. Length (Trucation)** | **Chunk-Size** | **Chunking** | **nDCG@10** |
| 192 | 192 | × | 14.45 |
| 8192 | 8192 | × | 36.81 |
| 8192 | 128 | ✓ | 47.99 |
| 8192 | 512 | ✓ | **48.34** |

To investigate whether chunking is beneficial for retrieval tasks when all texts are smaller than the maximum input token length of the model, we truncate texts to a fixed length and apply chunking afterwards. We then evaluate the retrieval performance with the same setup as in Section 4.1, using the `jina-embeddings-v2-small` model and applying all the non-synthetic retrieval tasks from the LongEmbed benchmark (Zhu et al., 2024). We use fixed-size chunking (see Section 4.1) with chunk sizes of 128 and 512 tokens. Table 5 demonstrates that retrieval with chunking significantly outperforms retrieval without chunking. The average relative improvement from chunking with 512 tokens is +24.47%. Only on the SummScreenFD task did retrieval without chunking perform slightly better. Furthermore, truncating at 8192 tokens generally performs better than truncating at 192 tokens, indicating that long-text embedding models still provide an advantage over embedding models with short context lengths.

## A.2  RETRIEVAL WITH OVERLAPPING CHUNKS

Table 6: Comparison of using Overlapping or Non-Overlapping Chunks.
Model: `jina-embeddings-v2-small`. the fixed-size strategy (256 tokens), optional overlap (16 tokens). Scores are in nDCG@10 [%]

| Dataset | Naive Chunking | | Late Chunking | |
|---|---|---|---|---|
| | w/ Overlap | w/o Overlap | w/ Overlap | w/o Overlap |
| SciFact | 64.2 | 61.7 | **66.1** | 65.9 |
| NFCorpus | 23.5 | 22.8 | 30.0 | **30.5** |
| FiQA | 33.3 | 32.8 | 33.8 | **34.0** |
| TRECCOVID | 63.4 | 64.5 | 64.7 | **64.9** |

In practice, engineers often construct overlapping chunks to prevent the loss of context at the chunk boundaries (Safjan, 2023). To analyze whether this improves retrieval performance with `jina-embeddings-v2-small` and the evaluation setup from Section 4.1, we ran the BeIR benchmark tasks using fixed-size chunking (256 tokens) both with and without an overlap of 16 tokens. The results in Table 6 do not show a clear advantage of using overlaps. The nDCG@10 scores are relatively similar regardless.

