# OpenReview forum: "Late Chunking: Contextual Chunk Embeddings Using Long-Context Embedding Models"
_ICLR.cc/2025/Conference — Submitted to ICLR 2025_

### Official Review · Reviewer_2G5K · 2024-10-30

**Soundness:** 2
**Presentation:** 3
**Contribution:** 3
**Rating:** 5
**Confidence:** 4

**Summary:**

This paper introduces a novel technique called "late chunking" for improving text embeddings in retrieval tasks by leveraging long-context embedding models. Unlike traditional chunking methods that split text before encoding, late chunking first encodes the entire document and then applies chunking, thereby preserving full contextual information within each chunk. This paper evaluates this approach on multiple retrieval datasets and demonstrates that late chunking consistently outperforms naive chunking methods across various chunking strategies (fixed-size, sentence-based, semantic) and models.

**Strengths:**

1. This paper proposes a late chunking technique that utilizes a long-context retriever to encode the full text before performing chunking, which reduces information loss caused by direct chunking.
2. This paper conducts extensive analytical experiments to explore the technical details of late chunking.

**Weaknesses:**

1. The experiments are not comprehensive enough. As only a subset of the BEIR benchmark is used in Section 4.1, limiting the assessment of the effectiveness of late chunking.
2. The proposed method needs high computational resources: Late chunking requires encoding the entire input with a long-context LLM before chunking, whereas standard chunking only encodes each chunk separately, resulting in shorter sequence lengths and reduced attention computation costs. As noted in Section 4.1, "splitting documents into smaller chunks increases the computational effort of the evaluation."
3. When dealing with longer texts, a sliding-window approach is still required, which could still lead to the loss of long-range dependency information.

**Questions:**

1. In Table 2, it seems that late chunking aims to better segment chunks, yet the use of sentence boundaries and fixed-size boundaries indicates that both late chunking and naive chunking methods are dividing chunks in the same way. Then why does late chunking still can generate higher-quality embeddings and achieve better performance?
2. Do the authors believe that late chunking could yield better results with LLM retrievers employing causal attention mechanisms?

---

> ### Author Response · Authors · 2024-11-15
>
> Firstly, thank you for your comments. We hope to clear up any misunderstandings so the contribution of our paper is more clear.
>
> Now we would like to answer your questions.
>
> > In Table 2, it seems that late chunking aims to better segment chunks, yet the use of sentence boundaries and fixed-size boundaries indicates that both late chunking and naive chunking methods are dividing chunks in the same way. Then why does late chunking still can generate higher-quality embeddings and achieve better performance?
>
> There might be a substantial misunderstanding of the late chunking idea we proposed in the paper. Late chunking itself does not modify the way texts are segmented. The modification that we propose is to apply the chunking after the whole text is encoded with the language model, i.e., we first obtain a sequence of token embeddings and then apply the chunking after that just during the mean pooling step that combines the token embeddings into a text embedding (for each chunk). So the use of late chunking is independent from the technique that is used to determine the segments and we primarily evaluate late chunking against chunking without our late chunking when using the same boundaries for the segments (with call the normal embedding method "naive chunking"). This process is described extensively throughout the paper, and a high level overview can be found in the introduction at the bottom of page 1. Our hypothesis for the improved performance of late chunking is that it can retain contextual interactions between chunks within the embeddings themselves, due to the embedding process happening first, on the unchunked text.
>
> > Do the authors believe that late chunking could yield better results with LLM retrievers employing causal attention mechanisms?
>
> Most embedding models rely on bi-directional attention. When training LLMs for a text embedding task, the attention mechanism is often changed from a causal to a bidirectional one. For example the two best LLM-based embedding models on the MTEB (Massive Text Embedding Benchmark): NV-Embed-v2 [1] and gte-Qwen2-7B-instruct (https://huggingface.co/Alibaba-NLP/gte-Qwen2-7B-instruct) make such a modification. Accordingly, we don't think that late chunking is particularly useful for causual attention. We don't have an educated guess whether late chunking would perform better with causual attention. However, this is beyond the scope of this paper, which is primarily focused on improving the representations of chunked text primarily for existing embedding models. This is an interesting direction for future research though, so thank you for raising this point.
>
> [1] Lee, Chankyu, et al. "NV-Embed: Improved Techniques for Training LLMs as Generalist Embedding Models." arXiv preprint arXiv:2405.17428 (2024).

---

> > ### Author Response · Authors · 2024-11-15
> >
> > We also have some remarks regarding the weaknesses pointed out:
> >
> > > The experiments are not comprehensive enough. As only a subset of the BEIR benchmark is used in Section 4.1, limiting the assessment of the effectiveness of late chunking.
> >
> > We think this critical point is not completely justified. On the one hand, we also evaluate on all datasets of the LongEmbed [2] benchmark as well as BeIR (see Figure 3 and Figure 4), chosen as we wanted to include long context documents to fully test the capabilties of late chunking in long range contextual dependencies. Moreover, we evaluated on most of the BeIR tasks and did not exclude tasks randomly. The only tasks that are excluded are tasks which are not available on the BeIR repository (https://github.com/beir-cellar/beir), datasets with > 1M documents as they don't allow such a comprehensive evaluation (all combinations of models, chunking strategies, and chunk sizes for some setups) due to the high computational costs, and CQADupstack as it is has a different structure and methodology for evaluation. We believe the combination of this and the LongEmbed dataset provides an extremely comprehensive set of real-world and synthetic data to test the method on.
> >
> > > The proposed method needs high computational resources: Late chunking requires encoding the entire input with a long-context LLM before chunking, whereas standard chunking only encodes each chunk separately, resulting in shorter sequence lengths and reduced attention computation costs. As noted in Section 4.1, "splitting documents into smaller chunks increases the computational effort of the evaluation."
> >
> > Firstly, we believe that there is a misunderstanding. That "splitting documents into smaller chunks increases the computational effort of the evaluation." (Section 4.1) is not a limitation of late chunking (our method proposed in the paper) but rather a problem of retrieval with small chunks in general. Indeed, the attention calculation is indeed more computationally expensive with increasing token length, however, in practice flash attention algorithms are usually used to make the attention with increasing token length increasingly more memory efficient (e.g. https://github.com/Dao-AILab/flash-attention/blob/main/assets/flashattn_memory.jpg) and faster (https://github.com/Dao-AILab/flash-attention/raw/main/assets/flash2_a100_fwd_bwd_benchmark.png). For more details, please see [3]. So whilst there is a small computational overhead compared to naive chunking, we do not believe it is significant for greater consideration nor does it scale significantly worse with modern attention architectures.
> >
> > > When dealing with longer texts, a sliding-window approach is still required, which could still lead to the loss of long-range dependency information.
> >
> > While our approach is not solving this problem, we want to point out that preserving dependencies beyond the maximum token length of the model is not the scope of this paper. Some of the models we used, e.g., jina-embeddings-v2-small allows exceeding its maximum token length and the use of relative positional encodings [4] allow extrapolation. Nevertheless, as our results demonstrate, long late chunking is still as effective compared to naive chunking as in setups with smaller chunks that do not exceed the maximum sequence length. You are correct that super long range dependency information can be excluded, but as a comparison, each chunk within late chunking has the capability to include context from ~10 pages of text, whereas naive chunking has the context from the chunk _only_, which is the main contribution of our methodology.
> >
> > We hope those explanations clarify your misunderstanding of our approach and its limitations. As we pointed out some of the weaknesses you mentioned are not related to the paper or might be caused by missing out parts of our paper. We also hope that any misunderstandings have been cleared up, and you are better able to appreciate the contribution of our work.
> >
> > [2] Zhu, Dawei, et al. "LongEmbed: Extending Embedding Models for Long Context Retrieval." arXiv preprint arXiv:2404.12096 (2024).
> >
> > [3] Dao, Tri. "FlashAttention-2: Faster Attention with Better Parallelism and Work Partitioning." ICLR 2024.
> >
> > [4] Press, Ofir, Noah A. Smith, and Mike Lewis. "Train short, test long: Attention with linear biases enables input length extrapolation." arXiv preprint arXiv:2108.12409 (2021).

---

> > ### Comment · Reviewer_2G5K · 2024-11-26
> >
> > Thank you for addressing my questions and concerns. I’d like to follow up on a few points from your response.
> >
> > How does late-chunking compare to other recent chunking methods, such as the ones discussed in:
> >
> > [1]: Dense X Retrieval: What Retrieval Granularity Should We Use?
> >
> > [2]: Landmark Embedding: A Chunking-Free Embedding Method for Retrieval-Augmented Long-Context Large Language Models

---

> > > ### Author Response · Authors · 2024-11-27
> > >
> > > We value your interest and are happy to address your follow-up points. The approach described [1] extracts propositions for each paragraph using an additional language model. It is related to the contextual embedding approach that we mentioned in the related work section. However, this one processes each paragraph independently, which might result in losing context across paragraphs. Compared to late chunking, it requires the use of an additional language model. Moreover, it cannot be used with any technique for segmenting the text (e.g. fixed-size, sentence-based, semantic chunking, ...) but is restricted to the texts produced by the model which limits it to applications that don't require embedding specific chunks. For example, if an application wants to highlight the relevant sentence, this does not work as the embedding does not correspond to a specific sentence, but rather to an output of the language model, which does not occur in the document in this form.
> > > The approach described in [2] is indeed related. It also aims to produce contextualized embedding representations. However, in contrast to late chunking, it trains an embedding model specifically to produce contextualized embedding representations of sentences. Late chunking does not require training a specific model and can be used with various techniques to segment text into chunks — it is not limited to sentences. We will add both references to the related work section.

---

### Official Review · Reviewer_sEvM · 2024-11-05

**Soundness:** 2
**Presentation:** 3
**Contribution:** 1
**Rating:** 3
**Confidence:** 5

**Summary:**

The paper propose a late-chunking strategy for text embeddings, where the texts are firstly past through a text encoder and then the pooling are done at chunks of the output token embeddings to form chunk embeddings. Experiment results show the proposed late chunking strategy performs better that naive chunking on the BEIR benchmark.

**Strengths:**

1. Chunking is an important problem in applying text embeddings in practical applications such as RAG.
2. The paper is clearly written and well presented.
3. The proposed solution is simple to implement for practitioners.

**Weaknesses:**

1. For naive chunking, the standard practice is to have some overlapping strides between chunks, and to include meta information such as document title in every chunks when available. It is unclear whether the author of this paper follows this practice in implementing the baselines.
2. The paper uses a relative small chunk size (up-to 512) in the experiments when the embeddings studied support 8k context length. As shown in the ablation, the gains from late chunking diminish when the chunk size goes from 16 up to 512. It is unclear whether it is still effective when the chunk size approaches the embedding length limit of 8k, where the benefit of chunking is most useful.

**Questions:**

NA

---

> ### Author Response · Authors · 2024-11-15
>
> Thank you for your review. We would like to address the two points you mentioned in the "Weaknesses" section of your review:
> > For naive chunking, the standard practice is to have some overlapping strides between chunks, and to include meta information such as document title in every chunks when available. It is unclear whether the author of this paper follows this practice in implementing the baselines.
> - We included the title (when available, as in NFCorpus and TRECCOVID) in the text, as this is the standard practice for evaluating these tasks. No additional meta-information is provided for the evaluation sets. Indeed, we did not evaluate chunking strategies that use overlapping chunks. Since there are numerous methods practitioners use for chunking, we could not cover all of them in our experiments. However, thank you for your suggestion, we have added an evaluation using overlapping chunks in Appendix A.2 of our revised submission. Since overlapping chunks is related to improving contextual dependencies between chunks, it is important to compare against. Overall, the results demonstrate no significant advantage of overlapping chunks for the BeIR benchmark tasks we evaluated. However, overlapping chunks did not reduce retrieval performance for naive and late chunking either.
>
> > The paper uses a relative small chunk size (up-to 512) in the experiments when the embeddings studied support 8k context length. As shown in the ablation, the gains from late chunking diminish when the chunk size goes from 16 up to 512. It is unclear whether it is still effective when the chunk size approaches the embedding length limit of 8k, where the benefit of chunking is most useful.
> - We completely disagree with the claim that chunking is mainly useful to handle cases where the texts reaches the maximum token length of the model, however, we acknowledge that we did not make this clear enough in our original submission. Therefore, we added some citations to previous works that have demonstrated that languange models in general [1] as well as embedding models in particular [2] cannot handle long text as well as short texts and using small chunk sizes is therefore more useful. In addition, we conducted another experiment (which is added to the updated submission in Appendix A.1) to demonstrate the limitations of long text embedding models and the advantage of (naive) chunking for retrieval applications. In particular, we show that even when only processing text which truncated (cut off) at the token limit of the model chunking performs on average ~24% better across all non-synthetic retrieval tasks in the LongEmbed benchmark [3].
>
> Furthermore, chunking has application outside of retrieval, such as in text classification for sentiment analysis [4], where chunking is necessary, independent to the length of the input document. This reference as well as a brief description has also been added to the revised paper.
>
> Given your low score of the paper, is there any other feedback you have aside from the two previously mentioned weaknesses that we could use to strengthen our work? We believe to have addressed your criticisms, and are therefore looking for other ways in which our research is deserving of the low score.
>
> [1] "LooGLE: Can Long-Context Language Models Understand Long Contexts?" by Jiaqi Li et al. (November 2023), https://arxiv.org/abs/2311.04939, shows that language models have problems to capture long dependencies
>
> [2] Zhou, Yuqi, et al. "Length-Induced Embedding Collapse in Transformer-based Models."
>
> [3] Zhu, Dawei, et al. "LongEmbed: Extending Embedding Models for Long Context Retrieval." arXiv preprint arXiv:2404.12096 (2024).
>
> [4] Patrick Lewis, Ethan Perez, Aleksandra Piktus, Fabio Petroni, Vladimir Karpukhin, Naman Goyal, Heinrich K¨uttler, Mike Lewis, Wen-tau Yih, Tim Rockt¨aschel, et al. Retrieval-Augmented Generation for Knowledge-Intensive NLP Tasks. Advances in Neural Information Processing Systems, 33:9459–9474, 2020.

---

### Official Review · Reviewer_uQcW · 2024-11-07

**Soundness:** 3
**Presentation:** 3
**Contribution:** 3
**Rating:** 6
**Confidence:** 4

**Summary:**

This paper proposed a novel method called “late chunking”, which leverages long context embedding models to first embed all tokens of the long text, with chunking applied after the transformer model and just before mean pooling. The resulting chunk embeddings capture the full contextual information, leading to superior results across various retrieval tasks. The method is generic enough to be applied to a wide range of long-context embedding models and works without additional training. To further increase the effectiveness of late
chunking, the authors also proposed a dedicated fine-tuning approach for embedding models. They experimented their method on BeIR benchmark and the results showed that by using late chunking, they are able to improve the retrieval performance (measured by NDCG) on several datasets.

**Strengths:**

- The proposed method is intuitive and simple but yield decent performance across embedding models and tasks.
- The proposed method can be directly used off the shelf, which would benefit the research community a lot.
- The paper is well organized and presented.

**Weaknesses:**

- I have some concerns regarding the results presented in Figure 3. In this figure, we observe that performance with late chunking declines across several datasets, including NarrativeQA (Chunk size > 128), 2WikiMultiHopQA (Chunk size > 16), SummScreenFD (Chunk size > 128), QMsum (Chunk size > 256), Needle-8192 (Chunk size > 4), and Passkey-8192 (Chunk size > 32). This pattern raises the question of whether the fusion of contextual information might actually lead to regression in fact-based retrieval tasks where extensive contextual information may be less relevant.

- I am also concerned about the experimental setup, particularly the choice of the BeIR benchmark as the primary testbed. The motivation for this choice feels less justified. To make a strong case that late chunking enhances retrieval performance in scenarios where contextual information is beneficial, it would be ideal to use a dedicated dataset (or a subset of datasets) where contextual information is  necessary for optimal retrieval performance. This approach would allow for a more informative breakdown of performance in contexts that benefit from contextual information versus those that do not. However, with the datasets selected, it is unclear to me how much contextual information contributes to performance gains and whether it might cause regressions in other scenarios.

- Another issue with the experimental setting is that only retrieval performance is measured not the downstream performance. Ultimately, downstream performance is what people care about. It is unclear whether improvements in retrieval performance translate into meaningful gains in downstream tasks.

- Section 4.5 feels somewhat incomplete. Rather than providing a systematic comparison, it functions more as a case study, which, in my opinion, adds less weight to the paper's central argument. I suggest reallocating this section’s space to address the concerns outlined above.

**Questions:**

In table 2, it's quite interesting to see the results show different trend on different datasets and different embedding models. For example, on TRECCOVID, late chunking helps least on Jv2 while on NFCorpus, it helps most on Jv2 and less on Jv3 and No. Do you have an idea what causes the differences?

---

> ### Author Response · Authors · 2024-11-15
>
> Firstly, thank you for your detailed review of our work. We hope that by addressing some of the points raised we can strengthen our paper, and make it more clear why our contribution is valuable.
>
> - There can indeed be small regressions when increasing sequence length, particularly in cases where only a small part of the document is relevant to the query. We added an additional experiment in Appendix A.1 of our updated submission. This experiment demonstrates that encoding large chunks of text into a single embedding representation performs poorly in such scenarios, which aligns with findings from previous works [1]. To mitigate this, we suggest using sufficiently small chunk sizes in these cases. In such scenarios, late chunking also performs well. As you noted, the regression for late chunking is particularly strong for the Needle and Passkey datasets. However, it is important to highlight that these datasets represent extreme cases as they are non-realistic, synthetic datasets constructed such that arbitrary, unrelated text surrounds a small amount of relevant text [2]. Consequently, the semantics of the chunk can indeed be obfuscated by late chunking. We do not claim that late chunking is strictly better than naive chunking across all scenarios, such as in Needle and Passkey, but across most real-world datasets it provides favourable results.
>
> - Our primary testbeds are two diverse sets of data, from BeIR and LongEmbed [2], which we believe represent a good portion of examples of different types of text. It would indeed strengthen the claim of late chunking if we created a specialised dataset to test it. However, the synthetic datasets in LongEmbed (needle and passkey) already represent corner cases where contextual information is not relevant, which we experiment on. In the paper, we prioritise experimenting on real world data to see the application of late chunking in retrieval tasks, and show that on this real data late chunking generally improves the performance of naive chunking across a range of scenarios.
>
> - While we agree that an evaluation on more downstream tasks would provide stronger motivation, we believe that retrieval itself has a lot of application. The new experiment in appendix A.1 shows that chunking enhance the retrieval performance on the non-synthetic retrieval tasks of the LongEmbed benchmark by ~24% in average and late chunking further improves it. Also, passage retrieval is in fact retrieval on chunks of documents.
>
> - Regarding your questions, jina-embeddings-v3 and the Nomic AI model both use rotary positional encodings while jina-embeddings-v2 uses AliBi. We could imagine that this has an influence here. However, we have no idea why they exhibit different trends specifically on those two datasets. Both NFCorpus and TRECCOVID contain documents from the medical domain.
> Initially, we found a strong correlation between the average length of the documents and the gains achieved by late chunking when using fixed-size chunking, however after evaluating more chunking strategies and models it wasn't that clear anymore. Therefore, we didn't mentioned it.
>
> [1] Zhou, Yuqi, et al. "Length-Induced Embedding Collapse in Transformer-based Models."
>
> [2] Zhu, Dawei, et al. "LongEmbed: Extending Embedding Models for Long Context Retrieval." arXiv preprint arXiv:2404.12096 (2024).

---

### Official Review · Reviewer_i4oQ · 2024-11-11

**Soundness:** 2
**Presentation:** 3
**Contribution:** 2
**Rating:** 5
**Confidence:** 4

**Summary:**

The paper introduces late chunking for document embeddings, which suggests that instead of chunking the text and then computing the embedding for individual chunks, one can alternatively first compute the embeddings for the whole document (or a great portion of it containing the desired chunk, in their long late chunking method), and then extract the embeddings for that chunk. Experiments conducted on retrieval tasks show the effectiveness of the proposed method and many ablations are conducted.

**Strengths:**

The experiments are well designed and the paper is easy to understand.
The performance of the method is consistently better than the compared baseline on almost all tasks & models experimented.

**Weaknesses:**

While the proposed method shows a good consistent, it seems to only work for an imaginary scenario --- Chunking methods are designed such that models can handle longer piece of text, but the proposed method only works if we can encode text longer than the chunk size.

**Questions:**

About the weakness, the reviewer can still imagine that in some cases where the chunk size is much smaller than the model length, this method can be useful. The author should present more practical examples and arguments that shows current practice often overlook this design, and that the work can signal the importance of late chunking.

---

> ### Author Response · Authors · 2024-11-15
>
> Thank you for the review. You are correct that the long context capabilities of embedding models are not enhanced by late chunking, but this is beyond the scope of this paper, where we seek to improve the ability of chunking given that this issue exists and chunking is therefore necessary. Additionally, we strongly disagree with the claim that chunking is primarily useful for handling cases where the text exceeds the model’s maximum token length. We acknowledge that we did not make this point clear enough in our submission. To address this, in our revised paper, we have added citations to prior work demonstrating that language models in general [1], as well as embedding models in particular [2], struggle to handle long texts as effectively as shorter ones.
>
> As shown in Figures 3 and 4 of the paper, the retrieval performance for late chunking is generally better when using smaller chunks compared to creating embeddings with the maximum allowable token length. Additionally, we conducted another experiment (now included in the updated submission in Appendix A.1) to further illustrate the limitations of long-text embedding models and the advantages of (naive) chunking in retrieval applications, highlighting the necessity for chunking longer texts, even when the text itself is below the token limit for the embedding model. In particular, we show that even when only processing text which truncated (cut off) at the token limit of the model chunking performs on average ~24% better across all non-synthetic retrieval tasks in the LongEmbed benchmark [3].
>
> Accordingly, we also disagree with the assertion that the retrieval of chunks, as performed in this paper, represents an imaginary scenario. On the contrary, chunking has been widely studied for retrieval systems such as RAG [4] and passage retrieval [5, 6], as well as text classification tasks such as sentiment analysis [7]. We have added these citations as well as a brief explanation to the revised paper, to better understand the application of our work.
>
> By adding extra details and citations to the applications of chunking, as well as the extra experiment in Appendix A.1, we believe this has strengthened our paper. We encourage you to look at the revised version. Please let us know if there is anything else concerning you about the contributions in our work.
>
> [1] "LooGLE: Can Long-Context Language Models Understand Long Contexts?" by Jiaqi Li et al. (November 2023), https://arxiv.org/abs/2311.04939, shows that language models have problems to capture long dependencies
>
> [2] Zhou, Yuqi, et al. "Length-Induced Embedding Collapse in Transformer-based Models."
>
> [3] Zhu, Dawei, et al. "LongEmbed: Extending Embedding Models for Long Context Retrieval." arXiv preprint arXiv:2404.12096 (2024).
>
> [4] Patrick Lewis, Ethan Perez, Aleksandra Piktus, Fabio Petroni, Vladimir Karpukhin, Naman Goyal, Heinrich K¨uttler, Mike Lewis, Wen-tau Yih, Tim Rockt¨aschel, et al. Retrieval-Augmented Generation for Knowledge-Intensive NLP Tasks. Advances in Neural Information Processing Systems, 33:9459–9474, 2020.
>
> [5] James P Callan. Passage-level evidence in document retrieval. In SIGIR’94: Proceedings of the Seventeenth Annual International ACM-SIGIR Conference on Research and Development in Information Retrieval, organised by Dublin City University, pp. 302–310. Springer, 1994.
>
> [6] Gerard Salton, James Allan, and Chris Buckley. Approaches to passage retrieval in full text information systems. In Proceedings of the 16th annual international ACM SIGIR conference on Research and development in information retrieval, pp. 49–58, 1993.
>
> [7] Dorottya Demszky, Dana Movshovitz-Attias, Jeongwoo Ko, Alan Cowen, Gaurav Nemade, and Sujith Ravi. Goemotions: A dataset of fine-grained emotions. In Proceedings of the 58th Annual Meeting of the Association for Computational Linguistics, pp. 4040–4054, 2020.

---

### Author Response · Authors · 2024-11-15
**Rebuttal Submission to Address Reviews**

We submitted a rebuttal to address the reviewers' comments. The primary critical point raised in multiple reviews was the claim that chunking is most useful when the text length exceeds the maximum token capacity of text embedding models. We argue that this is a misconception, as embedding models with high token limits often perform poorly on long texts [1,2]. In practical applications, chunking is frequently applied to much smaller text segments. However, we acknowledge that our original submission did not clearly emphasize this point. To address this, we added references to studies investigating this limitation, as well as papers demonstrating the use of chunking with relatively small chunk sizes. Additionally, we included a new experiment in Appendix A.1, showing that chunking improves retrieval performance significantly (on average ~24%) even for texts within the model's maximum token limit.
Furthermore, one reviewer noted that we did not consider overlapping chunks, a common approach to avoid losing context. To address this, we conducted another experiment (Appendix A.2), which demonstrates that overlapping chunks do not lead to significant improvements for the evaluated BeIR tasks. Additionally, our late chunking approach achieves similar performance gains. Finally, we identified a minor error in the pseudocode of Algorithm 2 (which we verified was not present in our implementation) and have corrected it.
[1] "LooGLE: Can Long-Context Language Models Understand Long Contexts?" by Jiaqi Li et al. (November 2023), https://arxiv.org/abs/2311.04939
[2] Zhou, Yuqi, et al. "Length-Induced Embedding Collapse in Transformer-based Models." arXiv preprint arXiv:2410.24200 (2024).

---

### Meta-Review · Area_Chair_1Pdr · 2024-12-20

**Metareview:**

The paper proposes a latent chunking approach for contextualized chunk representation.

Reviewers generally gave borderline or rejection scores. Several major concerns are related to experimental setups and lack of evaluation on downstream tasks. Even the reviewer who gave a borderline leaning positive score pointed out same major concern. I believe the paper does not meet the bar of ICLR.

**Additional Comments On Reviewer Discussion:**

Reviewers generally gave borderline or rejection scores. Even the reviewer who gave a borderline leaning positive score pointed out major concerns that the paper lacks evaluation in downstream tasks.

---

### Decision · Program_Chairs · 2025-01-22

Reject